# 3D Chromatin Organization Involving MEIS1 Factor in the *cis*-Regulatory Landscape of *GJB2*

**DOI:** 10.3390/ijms23136964

**Published:** 2022-06-23

**Authors:** Anaïs Le Nabec, Clara Blotas, Alinéor Briset, Mégane Collobert, Claude Férec, Stéphanie Moisan

**Affiliations:** 1University Brest, Inserm, EFS, UMR 1078, GGB, F-29200 Brest, France; clarablotas@yahoo.fr (C.B.); alienor.briset@etu.univ-lyon1.fr (A.B.); megane_collobert@hotmail.fr (M.C.); claude.ferec@gmail.com (C.F.); 2Laboratoire de Génétique Moléculaire et d’Histocompatibilité, CHRU Brest, UMR 1078, F-29200 Brest, France

**Keywords:** gene regulation, 3D chromatin organization, enhancer-promoter contacts, CRISPR, 4C, deafness, DFNB1, GJB2, MEIS1

## Abstract

The human genome is covered by 8% of candidate *cis*-regulatory elements. The identification of distal acting regulatory elements and an understanding of their action are crucial to determining their key role in gene expression. Disruptions of such regulatory elements and/or chromatin conformation are likely to play a critical role in human genetic diseases. Non-syndromic hearing loss (i.e., *DFNB1*) is mostly due to *GJB2* (*Gap Junction Beta 2*) variations and *DFNB1* large deletions. Although several *GJB2 cis*-regulatory elements (CREs) have been described, *GJB2* gene regulation remains not well understood. We investigated the endogenous effect of these CREs with CRISPR (clustered regularly interspaced short palindromic repeats) disruptions and observed *GJB2* expression. To decipher the *GJB2* regulatory landscape, we used the 4C-seq technique and defined new chromatin contacts inside the *DFNB1* locus, which permit DNA loops and long-range regulation. Moreover, through ChIP-PCR, we determined the involvement of the MEIS1 transcription factor in *GJB2* expression. Taken together, the results of our study enable us to describe the 3D *DFNB1* regulatory landscape.

## 1. Introduction

The ENCODE Project Consortium allows us to understand that human candidate *cis*-regulatory elements (cCREs) span over 8% of the human genome [1]. These cCREs need to be identified and characterized to understand how our genes are regulated [2,3,4,5]. Gasperini et al. described three steps to support a cCRE as a biological CRE with strong evidence: epigenetic genomic perturbation changes the *cis* expression, episome-based demonstration of activity, and enhancer-associated biochemical annotations [6]. CREs play an important role in gene regulation by controlling expression and creating a specific regulatory landscape [7].

Furthermore, 80% of deafness is of genetic origin and, in the majority of cases, is caused by non-syndromic hearing loss (NSHL) [8]. The *GJB2* (*Gap Junction Beta 2*) gene, which is located within the *DFNB1* locus on 13q12 (hg19, chr13:20,500,000–21,525,000), accounts for the majority of NSHL and deafness [8,9]. *GJB2* is a short gene encompassing two exons (193 pb and 2141 pb) separated by one intron (3179 pb) [10]. *GJB2* encodes a connexin 26 transmembrane protein (Cx26). Cx26 is expressed on supporting cells of cochlea and plays a role in cochlear homeostasis [11,12,13]. Additionally, the *GJB2* gene has ubiquitous expression; it is well expressed in the supporting cells of human cochlea, fibrocytes and mesenchymal cells in the lateral wall, and basal and intermediate cells of the stria vascularis. Moreover, *GJB2* is expressed on ferret airways and lung cells during development and persists throughout life in the lungs; this seems to be similar in humans [14]. As there is not yet a disposable inner ear cell line, we decided to use the primary human nasal epithelial cells (HNECs) and small airway epithelial cells (SAECs), which effectively express *GJB2* (http://dnase.genome.duke.edu/geneDetail.php?ensemblID=gjb2 (accessed on 25 April 2022)) (Figure 1A) [15].

In addition, large *DFNB1* deletions have been associated with *DFNB1* patients: del-920 kb [16], del-101 kb del(*GJB2*-D13S175) [17], del(*GJB6*-D13S1830) [18,19,20], del(*GJB6*-D13S1854) [21], del-131kb [22], del-179 kb [23], del-8 kb [24], and a deletion of 3 kb in one patient [25].

Some deletions impact only one other connexin gene, the *GJB6* gene, but studies have described that *DFNB1* deafness is not due to the *GJB6* deletion but the dysregulation of *GJB2* expression [26,27,28]. Thus, few publications have hypothesized that CREs in a long range of *GJB2* would be present in the deleted regions.

Following these observations, using chromosome conformation capture carbon copy (5C) analyses and luciferase assays, we described in 2019 the first *GJB2* CREs [15]. Based on these first results, we wanted to better understand the *cis*-regulatory landscape of *GJB2*.

In this work, we used a combination of the circular chromosome conformation capture technique (4C), derivate CRISPR-Cas9 (clustered regularly interspaced short palindromic repeats-associated protein 9) approaches, and the immunoprecipitation of chromatin to define and establish a *GJB2* regulatory landscape. Herein, we highlight novel chromatin contacts between CREs and the *GJB2* promoter and show that the recruitment of CCCTC-binding factors (CTCFs) enables the establishment of *DFNB1* 3D chromatin organization, ensuring the correct expression of *GJB2*, due to physical contacts with several transcription factors.

## 2. Results

### 2.1. CRISPR Analyses Confirm GJB2 Biological Enhancers

To characterize *GJB2* CREs in endogenous conditions, as recommended in [6], we applied derivative technologies from the CRISPR/Cas system: CRISPR interference (CRISPRi) and CRISPR activation (CRISPRa). These two techniques enable the study of gene regulation on a transitional basis. CRISPRi uses a dead Cas9 (dCas9) fused to Krüppel-associated box (KRAB) and with a single guide RNA (sgRNA) targeting site (Figure 1B). This complex can strongly repress genes by targeting CREs [29]. Regarding CRISPRa, we used a synergistic activation mediator system (SAM) with a dCas9 fused to VP64 and sgRNA presenting a tetraloop and stem loop 2 lead to recruit MS2 bacteriophage coat proteins (Figure 1C). Moreover, MS2 is associated with NF-kB trans-activating subunit p65 (p65) and heat shock factor protein 1 (HSF1) to form the MPH complex (Figure 1C). VP64 and p65 lead to recruit transcription factors and a chromatin remodeling complex [30]. The SAM system more efficiently activates genes compared to dCas9-VP64 [29,30].

With CRISPRi, we wanted to repress *GJB2* expression, for which we used our cell type of interest, SAECs. In contrast, with CRISPRa, we sought to overexpress the *GJB2* gene, and thus had to use a cell line that does not express *GJB2*. *GJB2* expression was controlled on SAECs and human embryonic kidney 293T (HEK293T) cells. HEK293T cells did not have basal *GJB2* expression (Figure 1A) and could, therefore, be selected for CRISPRa.

To validate the efficiency of these approaches, we first targeted the *GJB2* promoter. An sgRNA was designed on the *GJB2* promoter (Figure 1D) targeting the GC-box at −81 bp of the transcription start site (TSS) on non-template strands (Appendix A).

Our previous study [15] allowed us to describe several enhancers, notably, a stronger C3 enhancer of approximately −220 kb of the promoter, which increased *GJB2* expression by 27% Next, to investigate and better characterize this region of 1043 bp, we targeted it with four sgRNAs (2, 3, 4, and 5), among which one was on the non-template strand (Figure 1D) (Appendix A).

For CRISPRi, we used a plasmid, which expresses the sgRNA and the dCas9 combined with KRAB and is tagged by enhanced green fluorescent protein (EGFP). It is a lentivirus plasmid of 14 kb, and given that SAECs are difficult to transfect, we chose to use lentivirus transduction to obtain an efficient infection. We controlled each lentivirus transduction with EGFP expression (Appendix A). RT-qPCR was performed to determine *GJB2* expression. *GJB2* expression was totally repressed when the promoter was targeted. Indeed, by targeting GC-box, we prevented *GJB2* expression. Additionally, these results confirmed a good inhibition with this approach. When targeting our strong C3 enhancer region, *GJB2* expression decreased by 80% with sgRNA_2 and was repressed by 50–70% for the other sgRNAs (Figure 1E).

For the CRISPRa technique, HEK293T cells were transfected with three different plasmids. One expressed the sgRNA; the second, the dCas9-VP64; and the last, the MPH complex (Figure 1C). When the *GJB2* promoter was targeted (sgRNA_1), *GJB2* expression, measured by RT-qPCR, increased by 7000% (Figure 1F). However, when the C3 region was targeted, only sgRNA_2 allowed the increase of *GJB2* expression by 2.5% (Figure 1F).

Both CRISPRi and CRISPRa enabled us to dysregulate *GJB2* expression. Results correlated between these two approaches allowed us to confirm the C3 region as an important enhancer of *GJB2* in endogenous conditions.

### 2.2. MEIS1 Transcription Factor Contributes to GJB2 Expression

Due to previous CRISPR analyses, we were able to reduce and determine a minimal C3 region that plays the most important role in *GJB2* expression. Indeed, sgRNA_2 presents the best results for both techniques. By zooming in on sgRNA_2, we looked for predicted transcription factor binding sites thanks to the JASPAR database, and we detected MEIS and STAT5A binding sites in this region (Figure 2A).

To investigate whether MEIS and STAT5A factors are implicated in *GJB2* expression due to the binding C3 region, we analyzed their expression on SAECs (Figure 2B). Only MEIS factors were expressed on our cell type of interest (Figure 2B). The MEIS transcription factor family plays an important role in gene regulation and chromatin dynamics [31]. MEIS factors can interact with components of the transcriptional machinery for efficient gene expression [31,32]. Following these results, we focused on the MEIS factor, initially using MEIS1 because, on the JASPAR database, the MEIS factor family presents the same DNA binding motif, and we already have a MEIS1 antibody that worked. Thus, we performed chromatin immunoprecipitation (ChIP) to determine whether MEIS1 binds to the C3 enhancer region.

We performed MEIS1 ChIP-PCR in three replicates and detected an amplification of the target region when chromatin was immunoprecipitated with MEIS1 but not with IgG (Figure 2C and Appendix A).

We confirmed that MEIS1 binds to our C3 region and seems to be implicated in *GJB2* regulation.

### 2.3. DFNB1 Chromatin Organization

In order to identify and characterize the 3D chromatin contacts in the *DFNB1* locus in SAECs, we used 4C-seq (Figure 3). We generated at least three 4C-seq biological replicates. Using PeakC, we identified specific long-range interactions indicated by green peaks (Figure 3). PeakC provided coordinates (Appendix A) for a significant region, which encompassed quite a few restriction fragments. For each significant peak, the region extended from 10 kb to 33 kb. Therefore, we described each region according to its average distance from *GJB2* TSS (−20 kb, −220 kb, −290 kb or −625 kb).

We first studied *GJB2* chromatin interactions by using a restriction fragment viewpoint of the *GJB2* promoter and identified three significant contacts (Figure 3A). The peak at −220 kb of the *GJB2* promoter corresponded to the previously reported C region, which encompassed the C3 enhancer. Immediately after, the strongest peak appeared at approximately −290 kb of the *GJB2* promoter, which correlated with a CTCF site. Then, we identified a new region interacting with the *GBJ2* promoter with the last peak at approximately −625 kb, in the *XPO4* gene.

We also realized 4C-seq to study the chromatin interactions of the C3 region with the *DFNB1* locus. We wanted to confirm the C3-*GJB2* promoter’s interaction with both designs. The 4C-seq profiles showed three significant peaks from replicates, among which one corresponded to the *GJB2* promoter (Figure 3B). Upstream, we identified that the C3 region also interacted with another region located at −20 kb from *GJB2*. Interestingly, the last peak at −625 kb in the *XPO4* gene was in the same region, which was newly identified to interact with the *GJB2* promoter. Thus, these three regions, the *GJB2* promoter, C3 region, and this region at −625 kb from the *GJB2* TSS, were in close physical proximity.

### 2.4. Region at −625 kb of GJB2 Promoter Corresponds to an Insulator

To investigate the function of the new region in *XPO4*, we used gene reporter assays to analyze if this region corresponded to a *cis*-regulatory element or if it interacted with *GJB2* and C3 through the recruitment of CTCFs. The region at −625 kb, which interacted with the *GJB2* promoter and C3 enhancer, had a length of 23 kb and presented small H3K27ac marks and a CTCF site. Due to its size, we amplified and cloned two sub-regions according to H3K27ac marks (Figure 4A) into the pGL3-Basic Vector, in which luciferase reporter expression was driven by a 1541 bp *GJB2* promoter fragment. The different constructs were individually co-transfected into SAECs with a beta-galactosidase plasmid as a control for transfection efficiency. Firefly luciferase expression was measured after 48 h and was normalized against the *PGJB2* construct alone, which was set to 1. As shown in (Figure 4B), none of these regions showed any effect on *GJB2* expression.

In our previous study, we analyzed CTCF binding along the *DFNB1* locus through ChIP-qPCR and showed that CTCFs were recruited on site at −631 kb [15]. Thus, this region at −625 kb from *GJB2* seems to be an insulator due to the recruitment of CTCFs.

## 3. Discussion

In this study, we investigated the regulatory landscape of *GJB2*. Our results, with derivative technologies from the CRISPR/Cas system (CRISPRi and CRISPRa), show *GJB2* down- and up-regulation. Targeting C3, *GJB2* expression decreased by 80% or increased by 2.5% for CRISPRi and CRISPRa, respectively (Figure 1E,F). These results confirmed the C3 region as an enhancer of *GJB2* in endogenous conditions.

Although CRISPRa assays have produced significant results, they are less important than CRISPRi. The CRISPRa complex seems to be less efficient than dCas9-KRAB. The SAM system (CRISPRa) used here is highly efficient at targeting the promoter because it is close to transcriptional machinery, but the target enhancer loses strength [35,36]. To increase activation, we could use an alternative system: enhancer-targeting CRISPRa (enCRISPRa), which seems be more efficient for distal regulatory elements [35]. The enCRISPRa approach uses dCas9 fused to p300 and the sgRNA with two MS2 hairpins, which recruit the MCP-VP64 fusion protein. This technique appears to display more robust perturbations of enhancer activity and gene transcription [35].

Moreover, CRISPRi/a analyses enable us to define a critical region inside C3. Indeed, most conclusive results have been obtained with sgRNA_2. This reduced region correlates with several transcription factor sites. With ChIP-PCR, we confirmed MEIS1 binding in the C3 region (Figure 2C). MEIS1 seems to play a key role in gene expression and thus in *GJB2* expression. It would be useful to disrupt its binding to test its regulatory function on *GJB2*. In further investigations, it could be interesting to also study STAT5B expression and binding of others MEIS factors in our region.

From the 4C analyses, we detected several significant interaction peaks for P*GJB2* and C3 viewpoints within the *DFNB1* locus. P*GJB2* and C3 4C-seq profiles can be visualized on Genome Browser of University of California, Santa Cruz (UCSC) to gain a better understanding (Figure 5A). We aligned them with available HNECs data (Hi-C, H3K27ac). HNECs are very similar to SAECs, and both express *GJB2* [15]. P*GJB2* chromatin interactions are highlighted in green and C3 interactions in blue. The 4C-seq results allowed us to detect three significant interactions of the *GJB2* promoter within the *DFNB1* locus: at −220 kb (Appendix A), corresponding to CREs previously described [15]; at −290 kb, which correlated with CTCF binding sites; and a peak at −625 kb, which also overlapped with a CTCF site. In our previous study, we analyzed CTCF binding along the *DFNB1* locus through ChIP-qPCR. We showed that CTCFs were recruited at almost all predicted sites, one of which was at −298 kb and another at −631 kb. Thus, we detected a DNA loop at CTCF sites inside the promoter and at −631 kb, which permitted *GJB2* regulation by bringing *GJB2* CREs closer to the promoter.

Concerning the C3 region, 4C data enabled us to identify three interacting chromatin contacts across the *DFNB1* locus. In particular, the *GJB2* promoter (P) thus confirmed the promoter–enhancer interaction with all designs.

Moreover, the C3 viewpoint interacted with another region, upstream of the *GJB2* promoter (−20 kb). This region contained several transcription factor binding sites, in particular, the E1A binding protein of 300 kDa (EP300/p300) and the forkhead box protein A1/A2 (FOXA1/FOXA2) (Figure 5B). These factors are described as “pioneer” factors, which play key roles in chromatin remodeling [37]. FOXA1 has the ability to open silent chromatin and promote the activation of the target gene by helping another factor to bind to its *cis*-regulatory region [37,38]. The p300 protein and its homologue CREB protein (p300/CEBP) presented an acetyltransferase activity. CEBP/p300 plays a crucial role in transcription initiation [39]. With this evidence, we needed to investigate the roles of p300 and FOXA1/2 in *GJB2* regulation.

The third interacting peak was at −625 kb, as with the *GJB2* promoter. Thus, both the *GJB2* promoter and C3 enhancer were physically proximal to this region, which was probably bound by the CTCF factor (Figure 5C). Indeed, CTCFs are necessary for enhancer–promoter contacts, especially in long-range regulation [40]. Therefore, the *GJB2* promoter interactions with CTCF sites would involve a regulatory loop, allowing CREs to be closer to the promoter (Figure 5C).

Kubo et al. showed that CTCF-independent promoters were closer to the active enhancer (<50 kb), although CTCF-dependent promoters require CTCFs to create long-range contact with enhancers [40].

Finally, all these results allow us to define a 3D *DFNB1* regulation model, where CREs at −220 kb and −20 kb from *GJB2* TSS become closer to the promoter via regulatory chromatin loops due to CTCF recruitment at −631 kb, −298 kb, and 15 kb. We can thus imagine a large regulatory landscape among all these regions (Figure 5D).

Derivative technologies from the CRISPR/Cas system are essential to better understand and characterize CREs. We envisage the disruption of cCREs at −625 kb and −20 kb in the future in order to study its impact on *GJB2* expression and chromatin organization.

In this study, we identified novel chromatin contacts between CREs and the *GJB2* promoter; the recruitment of CTCF factors enabled the establishment of *DFNB1* 3D chromatin organization, ensuring the correct expression of *GJB2* due to physical contacts with several transcription factors.

These data provide valuable information on the basic insights of the regulation of this gene, and the description of *GJB2 cis*-regulatory elements may be important for gene-targeting constructs for therapy development. They also afford new targets for mutation screening in the diagnosis of hearing impairment. Indeed, they open a new field of research on genetic deafness and presbycusis, also known as age-related hearing loss, which are major public health issues.

## 4. Materials and Methods

### 4.1. Cell Culture

Small airway epithelial cells (SAECs) were purchased from LGC Standards and grown in airway epithelial cell medium from PromoCell (Heidelberg, Germany). HEK293T cells were grown in DMEM (Dulbecco’s modified Eagle medium) (Ozyme, Saint-Cyr l’Ecole, France)/10% FBS (Foetal Bovine Serum) (Eurobio, Les Ulis, France). Cells were grown on plastic at a liquid interface at 37 °C in 5% CO_2_ saturated humid air.

### 4.2. Circular Chromosome Conformation Capture (4C)

We generated 4C libraries from cultured cells, as described in [41]. Furthermore, 7.5 × 10^6^ cells were collected and fixed at 1.5% of fresh formaldehyde, incubated for 10 min at 20 °C, and stopped with 125 mM glycine for 5 min at 4 °C. Cell pellets were washed with 1 mL cold PBS. Supernatants were discarded, and cell pellets were resuspended with 1 mL freshly prepared cold cell lysis buffer (50 mM Tris-HCl pH 7.5, 0.5% NP−40, 1% Triton X-100, 150 mM NaCl, 5 mM EDTA, and 1X protease inhibitors), prepared using Milli-Q (final volume, 1 mL), and incubated for 20 min on ice. DpnII (200U, NEB, Massachusetts, USA) was used for the first digestion. T4 DNA ligase (Promega, Wisconsin, USA) was used for ligation with a ligation mix (660 mM Tris-HCl, pH 7.5, 50 mM MgCl_2_, 50 mM DTT, and 10 mM ATP) and prepared using Milli-Q (final volume, 700 µL). The ligated DNA was purified using Nucleomag PCR beads (as described in [41]). The 3C templates were digested using NlaIII (50U), ligated with a final concentration of 5 ng/µL, and incubated overnight at 16 °C. The 4C templates were purified using Nucleomag PCR beads (MACHEREY-NAGEL, Hoerdt, France) and eluted on 300 µL 5 mM Tris-HCl pH 8 and 150 µL Milli-Q.

The library was then subjected to PCR using the Expand™ Long Template PCR System (Roche, Penzberg, Germany) and the oligos designed for the P*GJB2* or C3 viewpoint (Appendix A). PCR amplicons were purified using Nucleomag PCR beads kit and subjected to next-generation sequencing with Illumina Miniseq using 75 bp single-end reads, and FastQ files were generated.

### 4.3. C-Seq Analysis

Data were analyzed with the pipeline Pipe4C version 1.1.4 (Krijger Peter, Utrecht, The Netherlands) [41].

Raw reads were aligned to the hg19 genome using Bowtie2 version 1.4. and sorted using SAMtools version 2.2.3. (Heng Li, Massachusetts, USA) Significant peaks were identified using PeakC script version 0.2 (de Wit Elzo, Amsterdam, Netherlands).

### 4.4. CRISPR Interference/Activation

#### 4.4.1. Design and Cloning of gRNA

The design of single guide RNA (sgRNA) was realized using the CRISPROR tool (http://crispor.tefor.net/ (accessed on 13 march 2020)) to target P*GJB2* and C3 enhancers. The list of sgRNA can be found in the Appendix A.

Plasmid pLV hU6-sgRNA hUbC-dCas9-KRAB-T2a-GFP (Addgene #71237) was used for CRISPRi, and lenti sgRNA(MS2)_zeo backbone plasmid (Addgene #61427) was used for CRISPRa. Plasmid pLV hU6-sgRNA hUbC-dCas9-KRAB-T2a-GFP contained gRNA, and dCas9 was fused to KRAB and GFP proteins. Lenti sgRNA(MS2)_zeo backbone plasmid contained just gRNA tape. Quick Ligase (NEB) was used to clone annealed oligos into vectors digested by BsmBI. Oligo insertions were verified by sequencing.

#### 4.4.2. Lentivirale Production for CRISPRi

Lentivirus particles were produced by transfecting HEK293T with 1.6 µg of the interest plasmid (#71237), 1.05 µg of psPAX2 plasmid (Addgene #12260), and 0.55 µg of pVSV-G plasmid packaging (Addgene #138479) using Lipofectamine™ 3000 Reagent (Invitrogen Cat No. L3000001, Waltham, MA, USA). One day before transduction, 3 × 10^5^ SAECs were seeded in 6-well plates. At 48 h after HEK293T transfection, lentivirus particles were purified and transduced in SAECs. At 48 h after transduction, the expression of GFP was controlled using a fluorescence microscope.

#### 4.4.3. Plasmid Transfection for CRISPRa

Next, 4.5 × 10^5^ HEK293T cells were seeded in 6-well plates. Transfections were performed 24 h later with three vectors: 800 ng of lenti sgRNA(MS2) zeo backbone plasmid, 800 ng of MS2-P65-HSF1 plasmid (Addgene #89308), and 800 ng of Lenti dCas9-VP64 plasmid (Addgene #61425) using transfection reagent TransIT-2020 (Mirus, Madison, WI, USA).

#### 4.4.4. RT-qPCR

At 48 h after transduction (CRISPRi) or transfection (CRISPRa), cells were collected by using trypsin (Lonza, Basel, Switzerland), and RNA was extracted with RNeasy Plus Mini Kit (QIAGEN, Hilden, Germany). RNA was reverse-transcribed in cDNA using the SuperScript IV First-Strand Synthesis System (Thermofisher, Waltham, Massachusetts, USA). Quantitative PCR was performed in triplicate using ONEGreen^®^ FAST QPCR PREMIX (Ozyme) in the Light Cycler 480 (Roche). Primer sequences for *GJB2* were forward primer 5′-TTCCTCCCGACGCAGAGCAA and reverse primer 5′-TCCTTTGCAGCCACAACGAGGAT, and for beta-actin, forward primer 5′-GTTGCTATCCAGGCTGTG and reverse primer 5′-CACTGTGTTGGCGTACAG.

### 4.5. Chromatin Immunoprecipitation

Chromatin immunoprecipitation (ChIP) was performed using the SimpleChIP^®^ Plus Enzymatic Chromatin IP Kit (Magnetic Beads) (Cell Signaling #9005). Briefly, 12 × 10^6^ cells were collected and fixed with 1.5% of fresh formaldehyde, incubated for 10 min at 20 °C, and stopped with 125 mM glycine for 5 min at 4 °C. Next, the SimpleChIP^®^ Plus Enzymatic Chromatin IP protocol was followed. The Adaptive Focused Acoustics™ (AFA) Technology from Covaris (Waltham, MA, USA) was used to sonicate chromatin for 4 min with 5% of sonication (75 Watts) two times. Chromatin was precleared with protein G agarose beads (Cell Signaling Technology, Waltham, MA, USA) for 2 h at 4 °C and immunoprecipitated with 8.5 μL of MEIS1-specific antibodies (Atlas Antibodies HPA056000), a negative control IgG antibody, or a positive control Histone H3 antibody (Cell Signaling Technology) overnight at 4 °C.

PCRs were performed in triplicate using the HotStarTaq Master Mix (QIAGEN), and the primer sequences used are listed in Appendix A.

All immunoprecipitations were carried out in triplicate, using different chromatin preparations.

### 4.6. Plasmid Constructs

All the cloning steps were performed using the “In fusion^®^” strategy (Takara Bio, Kusatsu, Japan). Using the pGL3-Basic Vector (Promega), the 5′-flanking region of the *GJB2* gene (1541 bp, “P*GJB2*”) was cloned upstream from the firefly luciferase cDNA, at the HindIII site. Candidate CREs from the −625 kb site (−625-L and −625-R) were amplified and inserted downstream in the P*GJB2* construct at the SalI site. All the inserted fragments were verified by sequencing. The PCR primers used to amplify the *GJB2* promoter candidate CRE sequences are shown in Appendix A.

### 4.7. Luciferase Assays

Next, 1.25 × 10^5^ SAECs were seeded in 12-well plates. Transfections were performed 24 h later with the TransIT-2020 reagent (Mirus). An amount of 400 ng of each construct (P*GJB2*, −625-L, and −625-R) and 100 ng of a pCMV-LacZ construct, as an internal control, were used for each condition. Every condition was performed in triplicate. At 48 h post-transfection, the cells were washed once with 1X PBS and lysed with passive lysis buffer (Promega). Cell lysates were clarified by centrifugation at 12,000× *g* for 5 min at 4 °C. An amount of 20 µL of each protein extract was used to assay the luciferase activity and 25 µL for beta-galactosidase activity. We used Promega reagents and a multiwell plate reader Varioscan (Thermo Fisher). Results were presented as the relative luciferase activity, with the P*GJB2* construct activity equal to 1. *T*-tests were realized according to the variance results between each group, with Bonferroni correction.

### 4.8. Accession Numbers

The 4C-seq data from this study were deposited in the Gene Expression Omnibus (GEO) under accession: GSE201205.

## Figures and Tables

**Figure 1 ijms-23-06964-f001:**
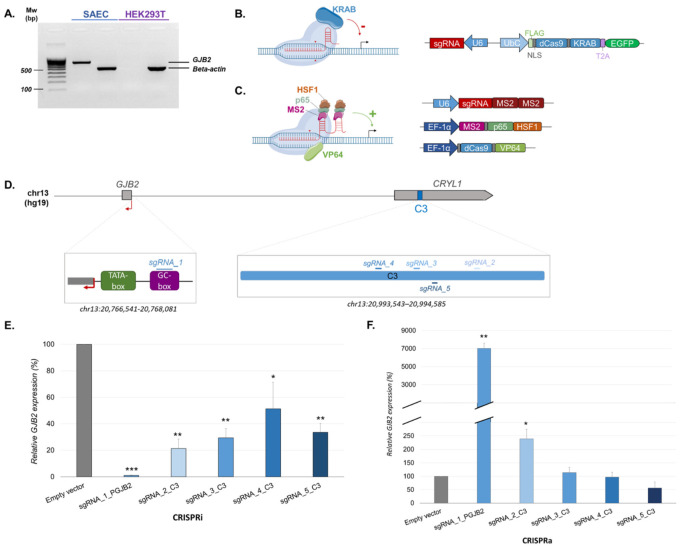
CRISPR analyses confirm *GJB2* biological enhancers. (**A**) *GJB2* expression in two cell lines: SAECs and HEK293T. (**B**) CRISPRi strategy for site specific targeting of the transcriptional repressor (Krüppel-associated box, KRAB). (**C**) CRISPRa strategy for site specific targeting of the transcriptional activators: VP64-MS2-p65 and HSF1. (**D**) Design of single guide RNA (sgRNA) target sites in *GJB2* promoter and C3 enhancers. (**E**) *GJB2* expression was quantified using RT-qPCR according to each gRNA of CRISPRi. Expression levels were normalized based on SAECs expressing no sgRNA (empty vector); *n* = 3, three independent lentiviral infections. (**F**) *GJB2* expression was quantified using RT-qPCR according to each gRNA of CRISPRa. Expression levels were normalized based on HEK293T cells expressing no sgRNA (empty vector); *n* = 4, four independent transfections. Variance and Student’s *t*-test with Bonferroni correction for multiple hypotheses were used to assess statistical significance. Error bars represent SEM: * <0.004; ** <0.0004; *** <0.0005.

**Figure 2 ijms-23-06964-f002:**
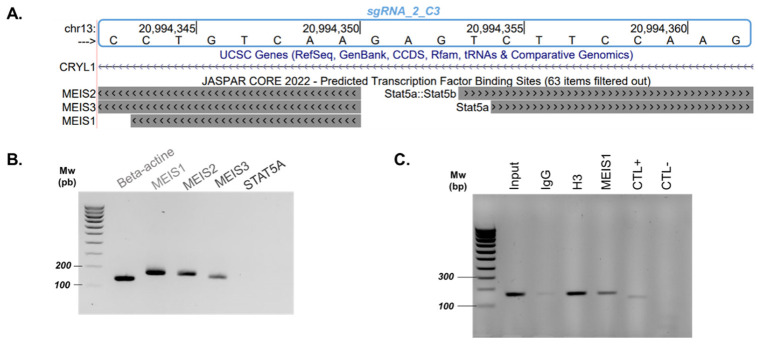
MEIS1 transcription factor contributes to *GJB2* expression. (**A**) Predicted transcription factor binding sites (JASPAR CORE 2022) on Genome Browser on University of California, Santa Cruz (UCSC) (hg19) for sgRNA_2 of C3 region. (**B**) Endpoint RT-PCR analysis of MEIS1, MEIS2, MEIS3, and STAT5a expression on SAECs (Appendix A). (**C**) Binding of MEIS1 on C3 enhancer by ChIP-PCR; *n* = 3, three independent chromatin immunoprecipitations. We used a negative control (CTL−) that had already been described as a CTCF binding region [33], and a positive control (CTL+), which was described in [34]. The results of one biological replicate are shown. Full gel pictures and data from two additional replicates can be found in Appendix A.

**Figure 3 ijms-23-06964-f003:**
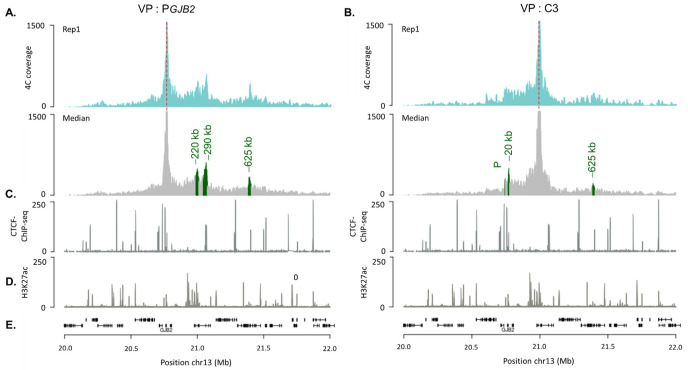
Chromatin contacts within the *DFNB1* locus. (**A**) 4C-seq profiles with *GJB2* promoter viewpoint (**A**) or C3 enhancer viewpoint (**B**) for one biological replicate; red line is the *GJB2* promoter or C3 viewpoint. Two additional replicates can be found in Appendix A. The following plot corresponds to contact profiles averaged over three biological replicates; statistically significant 4C-peaks are highlighted in green. (**C**) CTCF ChIP-seq on SAECs. (**D**) H3K27ac profiles on HNEC cell lines. (**E**) Genomic location of genes on *DFNB1* locus (hg19).

**Figure 4 ijms-23-06964-f004:**
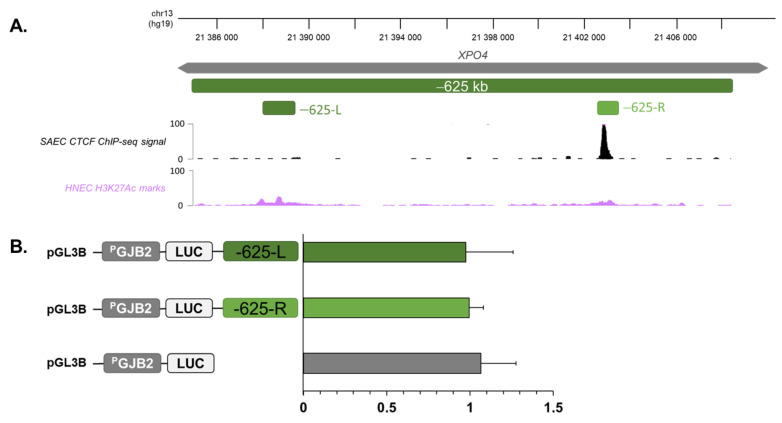
Region at −625 kb of *GJB2* promoter corresponds to an insulator. (**A**) Subsections of the −625 kb region into two candidate regions depending on their overlaps with H3K27ac marks from HNEC data. (**B**) SAECs were transfected with pGL3B luciferase reporter constructs containing the *GJB2* basal promoter (P*GJB2*; 1541 bp) and two cCREs (−625-L and −625-R). Luciferase data are shown relative to the *GJB2* basal promoter vector (=1). Error bars represent SEM (*n* = 9).

**Figure 5 ijms-23-06964-f005:**
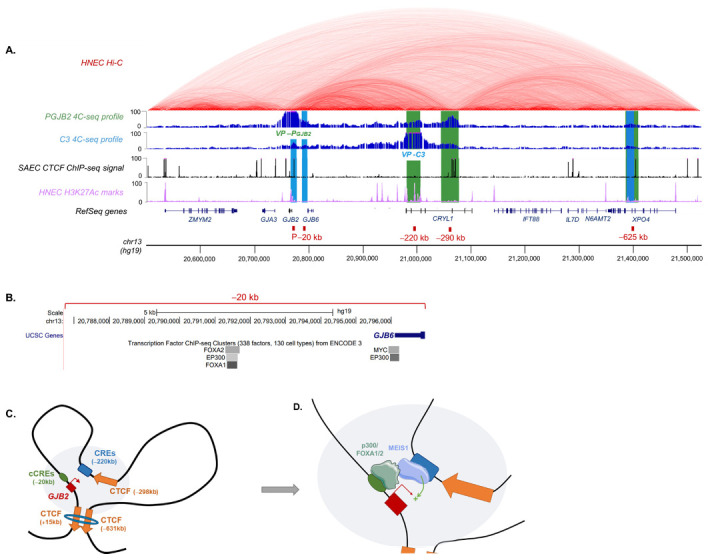
*GJB2* regulatory landscape. (**A**) UCSC visualization of chromatin interactions within *DFNB1* locus. Alignment of HNEC Hi-C data from Rao et al. 2014, 4C-seq profiles and significant interaction peaks (green for P*GJB2* and blue for C3), SAEC CTCF Chip-seq signal, HNEC H3K27ac marks, RefSeq genes, and genomic location of *DFNB1* locus (hg19). (**B**) Predicted transcription factor binding sites (ENCODE 3) on Genome Browser UCSC (hg19) for C3 4C peak at −20 kb of *GJB2* TSS. (**C**) 3D *DFNB1* regulation model. DNA loop extrusion by CTCF binding at +15 and −631 kb of *GJB2* promoter. Orange arrows indicate CTCF binding sites. Blue ring corresponds to cohesin. Grey circle indicates *GJB2* regulatory landscape. (**D**) *GJB2* regulatory landscape seems to require MEIS1, FOXA1/2, and p300. MEIS1 binds to C3 enhancer. FOXA1/2 and p300 could bind to the region upstream of *GJB2* promoter to recruit MEIS1 and activate transcription.

## Data Availability

The accession number for the raw reads and processed files for our 4C-seq datasets reported in this paper is GEO: https://www.ncbi.nlm.nih.gov/geo/query/acc.cgi?acc=GSE201205 (accessed on 21 April 2022).

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
