# Peer review of "3D Chromatin Organization Involving MEIS1 Factor in the cis-Regulatory Landscape of GJB2"

_ijms, 2022, doi:10.3390/ijms23136964_

Round 1

Reviewer 1 Report

This article investigates a gene associated with hearing loss/deafness in humans, GJB6. Previous 5C experiments by this group had shown the first CREs identified to interact with GJB6. This study further explored the regulation of GJB6 by CREs using 4C technologies and CRISPRa/i experiments. A new CRE is identified and is suggested to play a role as an insulator for GJB6 expression. A regulatory model including TFs and CREs is proposed. The paper has a story which could be followed, though I would recommend having a native English speaker go through and edit the grammar of the overall article. While the article could be understood, it was a struggle in some parts and would benefit from editing by someone who can improve the grammar throughout. Below are a few other suggestions I have to help improve the article.

2.1

First paragraph of results can be moved to introduction.

Line 115, cannot say that RNA Pol II recruitment was blocked unless you have performed a ChIP experiment for RNA Pol II to confirm this is the case. Is this possible?

Lines 117-118, either use % or fold when describing decreased expression i.e. say “... and is repressed by 50-75% for the other sgRNAs”.

Why do you imagine all of the sgRNAs were effective in the CRISPRi experiment, but only sgRNA_2_C3 was effective for the CRISPRa experiment? Did you examine TF binding sites on the regions where the other sgRNAs bind as well and find any similarities/differences?

2.2

Line 134, Figure 2B does not show RT-PCR results. This is a PCR product that has been run on a gel. Was RT-PCR run as well? If not, this should be reworded to reflect what was actually done.

Are there other cell lines available that have GJB2 expression that also express STAT5A? This should be considered a future direction.

Are there antibodies available to do MEIS2/3 ChIPs as well? How do MEIS1-3 differ from one another, both in structure and function?

2.3

Figure 3D, no mention is made anywhere in the article as to what is being indicated by this H3K27ac track, other than that HNEC cells are similar to SAEC cells. As it is an active transcription mark, it appears to show high active transcription of the C3 region. The GJB2 region also has active mark peaks, though not nearly to the same extent as C3. Any thoughts as to why this may be?

2.4

May wish to look at CRISPRa/i experiments on XPO4 in future, and comment on what might be expected as results from this experiment.

Author Response

Response: We agree with the Reviewer, we used editing services of MPDI to correct English grammar and orthography.

2.1

First paragraph of results can be moved to introduction.

Response: Yes, indeed, we moved the paragraph to line 40.

Line 115, cannot say that RNA Pol II recruitment was blocked unless you have performed a ChIP experiment for RNA Pol II to confirm this is the case. Is this possible?  

Response: You are right, thank you. We modified this sentence.

Lines 117-118, either use % or fold when describing decreased expression i.e. say “... and is repressed by 50-75% for the other sgRNAs”.  

Response: It is correct, we modified.

Why do you imagine all of the sgRNAs were effective in the CRISPRi experiment, but only sgRNA_2_C3 was effective for the CRISPRa experiment? Did you examine TF binding sites on the regions where the other sgRNAs bind as well and find any similarities/differences?  

Response: For CRISPRi, we bring a repressor complex which can be sufficient to repress while CRISPRa need to recruit others factor. Moreover, for CRISPRa we use a cell type which did not have GJB2 basal expression, as a consequence may be not have transcription factor which control GJB2 expression. Moreover, CRISPRa seems be less efficient to target cis-regulatory elements at distance.

For the second question, we verified the binding of transcription factors for the others sgRNA on JASPAR data base (2022) and there are lot of binding sites: NFIC, NFIX, TCF21, SCRT1, SCRT2, MXI1, ATOH7, TFAP4, WT1, ETV2, FOXO1, ELF1, ISL2, BATF, JUN, ZNF549, NKX3-1, FOSL2, BNC2, ATF3.

2.2

Line 134, Figure 2B does not show RT-PCR results. This is a PCR product that has been run on a gel. Was RT-PCR run as well? If not, this should be reworded to reflect what was actually done.

Response: We apologize for any confusion for the reviewer that shows we had not made this point very clear. We changed the description of this approach in the manuscript 

Are there other cell lines available that have GJB2 expression that also express STAT5A? This should be considered a future direction.

Response: Indeed, thank you, we do not think about it. We will consider this possibility.

Are there antibodies available to do MEIS2/3 ChIPs as well? How do MEIS1-3 differ from one another, both in structure and function?

Response: Yes, antibodies for MEIS2/3 exist. We used MEIS1 antibody because we knew that it works well and we already have it. Moreover, on JASPAR database, MEIS1, MEIS2 and MEIS3 have the same motif of binding.

2.3

Figure 3D, no mention is made anywhere in the article as to what is being indicated by this H3K27ac track, other than that HNEC cells are similar to SAEC cells. As it is an active transcription mark, it appears to show high active transcription of the C3 region. The GJB2 region also has active mark peaks, though not nearly to the same extent as C3. Any thoughts as to why this may be?

Response: We agree with the Referee and apologize for not indicating it. We modified the text to include the following sentences and references (lines 45-48): “As, there is not yet disposable inner ear cell line, we decided to use primary human nasal epithelial cell (HNEC) and small airway epithelial cells (SAEC) which well express GJB2 (http://dnase.genome.duke.edu/geneDetail.php?ensemblID=gjb2) (Figure 1A) [15].”

On UCSC, there is no H3K27ac track for SAEC.

Promoters can present actives marks like H3K27ac but are preferentially found on enhancer. Moreover, promoters can present others marks like H3K4me2/3 so that can explain these observations for GJB2 promoter.

2.4

May wish to look at CRISPRa/i experiments on XPO4 in future, and comment on what might be expected as results from this experiment.

Response: Indeed, it is a good proposition and very interesting to realize it. We add a sentence on discussion on line 290-291 “We envisage the disruption of cCREs, at -625 kb and -20 kb, in the future, to study its impact on GJB2 expression and chromatin organization.“

Reviewer 2 Report

In this study, Le Nabec and colleagues aimed to characterize the enhancer landscape and uncover factors regulating the expression of GJB2, which is involved in hearing loss.

The study was focused on a candidate enhancer locus named C3, which was previously identified by the group. To better characterize this C3 enhancer, CRISPRa CRISPRi were targeted to four loci spanning ~300 bp at the enhancer.  One of the sgRNA sequences contained MEIS binding motif and MEIS1 binding was tested by ChIP-qPCR.  Reciprocal 4C-seq revealed a chromosomal association between GJB2 gene and the 220kb distant C3 enhancer, and additional contact, 625kb from GJB2 gene.  The authors suggest that this locus serves as an insulator.

The data in this study is overall descriptive and requires major revisions to warrant publication.

Major:

~300bp region in the ~1000bp C3 enhancer was targeted by four different sgRNAs (2-5) in CRISPRi and CRISPRa experiments for assessing its potential regulatory role.  While sgRNA_2 showed a considerable activating and repressing capacity, sgRNA_3,4,5 showed only activating capacity.  This variability in the activity of very proximal sgRNAs targets could be technical or biological. In addition, the enhancer region is not limited to the 20bp of the sgRNA_2 sequence. Nevertheless, the authors identified binding motifs of MEIS and STAT5A transcription factors on the 20 bps of the sgRNA_2  sequence.  

Semi-quantitative RT-PCR indicated that all 3 members of the MEIS family of transcription factors are expressed in SAEC cells and the authors decided to focus on MEIS1 without explaining why. STAT5A was not detected in the Semi-quantitative RT-PCR but it is not clear why other members of the STAT family that recognize the same motif were not tested. In addition, I could not find the primer sequences that were used to quantify the expression of MEIS factors.

The authors need to explain why MEIS rather than STAT factors were selected for further analysis. The proposed explanation (L135-137) “MEIS transcription factors family plays an important role on gene regulation and chromatin dynamics. MEIS factors can interact with components of the transcriptional machinery for efficient gene expression.” Is rather general, and not referenced. It is okay to exclude STATs from further analysis but the choice in MEIS has to be justified.

MEIS1 binding on the C3 enhancer was tested by ChIP-PCR. To argue for MEIS1 binding, the signal at the C3 locus must be normalized to a negative control locus (a region that MEIS1 does not bind) in all the ChIP experiments and positive control (a region where MEIS1 is expected to bind) must be shown.  Notably- the depiction of MEIS1 binding does not indicate its function. MEIS1 regulatory function can be tested by its perturbation.

The 4C-seq experiments support the claim of a long-range association between GJB2 gene with the C3 region, as well as with the 625kb distant locus.

However, the reporter experiments and genomic data in figure 4 do not support that the 625kb distant locus serves as an insulator.

Minor:

Figures- overall the text is too small to read

Figure 1D- genomic coordinates should be added.

Figure 1F- graphs need titles.

Why two different MEIS1 forward primers are described in the ChIP method?

Some data that seems trivial (for example Figure 1B, C, E) could be moved to supplementary data.  Figure 3 can show only the averaged profiles, leaving the replicas in supplementary data.

Some of the results are overstated. For example, what is the basis to state that C3 region is a “key” enhancer of GJB2(L125)?  Although the abstract stated that “we determined involvement of several transcription factors on GJB2 expression.” only MEIS1 was tested.

The manuscript requires extensive English editing. Just to name few examples:  

Not clear- We better investigated the effect of CRISPR perturbation on GJB2 expression by targeting these CREs.

we realized??  chromatin immunoprecipitation (ChIP)

GJB2 gene regulation was not well understood

Proper citations are missing in multiple locations. For example, L135: “MEIS transcription factors family plays an important role on gene regulation and chromatin dynamics.  MEIS factors can interact with components of the transcriptional machinery for efficient gene expression.”

Author Response

In this study, Le Nabec and colleagues aimed to characterize the enhancer landscape and uncover factors regulating the expression of GJB2, which is involved in hearing loss.

The study was focused on a candidate enhancer locus named C3, which was previously identified by the group. To better characterize this C3 enhancer, CRISPRa CRISPRi were targeted to four loci spanning ~300 bp at the enhancer.  One of the sgRNA sequences contained MEIS binding motif and MEIS1 binding was tested by ChIP-qPCR.  Reciprocal 4C-seq revealed a chromosomal association between GJB2 gene and the 220kb distant C3 enhancer, and additional contact, 625kb from GJB2 gene.  The authors suggest that this locus serves as an insulator.

The data in this study is overall descriptive and requires major revisions to warrant publication.

Response: First of all, thanks to reviewer for the comments and suggestions that we are sure that will improve our manuscript.

Major:

~300bp region in the ~1000bp C3 enhancer was targeted by four different sgRNAs (2-5) in CRISPRi and CRISPRa experiments for assessing its potential regulatory role.  While sgRNA_2 showed a considerable activating and repressing capacity, sgRNA_3,4,5 showed only activating capacity.  This variability in the activity of very proximal sgRNAs targets could be technical or biological. In addition, the enhancer region is not limited to the 20bp of the sgRNA_2 sequence. Nevertheless, the authors identified binding motifs of MEIS and STAT5A transcription factors on the 20 bps of the sgRNA_2  sequence.

Semi-quantitative RT-PCR indicated that all 3 members of the MEIS family of transcription factors are expressed in SAEC cells and the authors decided to focus on MEIS1 without explaining why.  Response: We focused on MEIS1 binding because we knew that our MEIS1 antibody works well and we already have it. Moreover, on JASPAR database, MEIS1, MEIS2 and MEIS3 have the same motif of binding.

STAT5A was not detected in the Semi-quantitative RT-PCR but it is not clear why other members of the STAT family that recognize the same motif were not tested.

Response: It is correct we did not verify the expression of the other members of STAT family. But it is a good remark and we will take it account.

In addition, I could not find the primer sequences that were used to quantify the expression of MEIS factors. 

Response: We agree with the Referee and apologize for not indicating it. We added primer sequences on Table S2.

The authors need to explain why MEIS rather than STAT factors were selected for further analysis. The proposed explanation (L135-137) “MEIS transcription factors family plays an important role on gene regulation and chromatin dynamics. MEIS factors can interact with components of the transcriptional machinery for efficient gene expression.” Is rather general, and not referenced. It is okay to exclude STATs from further analysis but the choice in MEIS has to be justified.

Response: The reviewer is right, we missed to referenced these sentences. We add the reference on the text and on the references paragraph “The MEIS transcription factor family plays an important role in gene regulation and chromatin dynamics [31]. MEIS factors can interact with components of the transcriptional machinery for efficient gene expression [31,32].”

MEIS1 binding on the C3 enhancer was tested by ChIP-PCR. To argue for MEIS1 binding, the signal at the C3 locus must be normalized to a negative control locus (a region that MEIS1 does not bind) in all the ChIP experiments and positive control (a region where MEIS1 is expected to bind) must be shown. Notably- the depiction of MEIS1 binding does not indicate its function. MEIS1 regulatory function can be tested by its perturbation.

Response: Reviewer comment is correct and very relevant, we analysed two other regions (positive and negative controls) and we modified the legend figure (Lines 166-167). It is correct, we can not affirm that MEIS1 is implicated on GJB2 regulation, we just can speculate by its binding. We accordingly, we modified the text: “We confirmed that MEIS1 binds to our C3 region, and seems to be implicated in GJB2 regulation. could regulate GJB2 expression by this interaction.”

And we modify on discussion (lines 235-237): “MEIS1 seems to play a key role in gene expression and thus in for GJB2 expression. It would be useful to disrupt its binding to test its regulatory function on GJB2.” 

The 4C-seq experiments support the claim of a long-range association between GJB2 gene with the C3 region, as well as with the 625kb distant locus. However, the reporter experiments and genomic data in figure 4 do not support that the 625kb distant locus serves as an insulator.

Response: Indeed, we just can speculate that -625kb region could be serves as an insulator.

Minor:

Figures- overall the text is too small to read.

Response: We modified the figures: text sizes are increased.

Figure 1D- genomic coordinates should be added.

Response: We added the genomic coordinates.

Figure 1F- graphs need titles.

Response: We added a title on the bottom of figure 1F.

Why two different MEIS1 forward primers are described in the ChIP method?

Response: We modified the primers (Table S5) and we added primer sequences for positive and negative controls.

Some data that seems trivial (for example Figure 1B, C, E) could be moved to supplementary data.  Figure 3 can show only the averaged profiles, leaving the replicas in supplementary data.

Response: Indeed, it is a good recommendation. We conserved just one replicate and the median profile on Figure 3. We added the replicates of 4C-seq profiles on Figure S3.

Some of the results are overstated. For example, what is the basis to state that C3 region is a “key” enhancer of GJB2(L125) Although the abstract stated that “we determined involvement of several transcription factors on GJB2 expression.” only MEIS1 was tested.

Response: We modified the sentences: lines 137-139 “Results correlate between these two approaches and allowed to confirm C3 region as key an important enhancer of GJB2 in endogenous condition” and line 20-21: ”Moreover, with ChIP-PCR we determined involvement of MEIS1 several transcription factors on GJB2 expression.”

The manuscript requires extensive English editing. Just to name few examples:  

Not clear- We better investigated the effect of CRISPR perturbation on GJB2 expression by targeting these CREs.

we realized??  chromatin immunoprecipitation (ChIP)

GJB2 gene regulation was not well understood

Proper citations are missing in multiple locations. For example, L135: “MEIS transcription factors family plays an important role on gene regulation and chromatin dynamics.  MEIS factors can interact with components of the transcriptional machinery for efficient gene expression.”

Response: We understand well this remark, and we take it account. We corrected the English grammar and syntax with MPDI service.

Reviewer 3 Report

Mutations in Gap Junction Beta 2 (GJB2 or DFNB1) have been reported to be a major cause of non-syndromic hearing loss worldwide. In the present study, the authors have demonstrated cis regulatory regions of GJB2 and associated transcription factors. To accomplish the aim of the study, 4C-seq technique was utilized to define chromatin contacts and long-range regulation information; and ChIP-PCR to identify transcription factors to corresponding regulatory regions.

The article is well written and may be accepted. However there are few minor points that may be addressed.

1. In the abstract the sentence, "Non-syndromic hearing loss (i.e. DFNB1) is 14 mostly due to GJB2 variations and DFNB1 large deletions", is confusing. GJB2 gene synonym is DFNB1. The sentence needs to be rephrased for better clarity.

2. Figure 2B depicts binding of MEIS1, 2, and 3 binding into the C3 regions. The author may explain the rational behind differential binding of the transcription factors.

3. The authors have shown chromatin interactions within the DFNB1 locus. The authors may provide TAD coordinates for the DFNB1 locus (Figure 1A).

Author Response

Mutations in Gap Junction Beta 2 (GJB2 or DFNB1) have been reported to be a major cause of non-syndromic hearing loss worldwide. In the present study, the authors have demonstrated cis regulatory regions of GJB2 and associated transcription factors. To accomplish the aim of the study, 4C-seq technique was utilized to define chromatin contacts and long-range regulation information; and ChIP-PCR to identify transcription factors to corresponding regulatory regions.

The article is well written and may be accepted. However there are few minor points that may be addressed.

Response: Thank you a lot for this kind return. We tried to answer for each point and we hope that with these corrections we will improve our paper.

  1. In the abstract the sentence, "Non-syndromic hearing loss (i.e. DFNB1) is 14 mostly due to GJB2 variations and DFNB1 large deletions", is confusing. GJB2 gene synonym is DFNB1. The sentence needs to be rephrased for better clarity.

Response: Indeed, we clarified the sentence: lines 14-15 “Non-syndromic hearing loss (i.e., DFNB1) is mostly due to GJB2 variations and DFNB1 large deletions, which remote the gene or cis-regulatory elements (CREs).”

  1. Figure 2B depicts binding of MEIS1, 2, and 3 binding into the C3 regions. The author may explain the rational behind differential binding of the transcription factors.

Response: We used the same cDNA quantity for the PCR and from the same cDNA, we purposed that differential binding of the transcription factors could be due to PCR efficiency. It could be useful to realize qPCR to determine if there are differential expression of this transcription factors on SAEC cells.

  1. The authors have shown chromatin interactions within the DFNB1 locus. The authors may provide TAD coordinates for the DFNB1 locus (Figure 1A).

Response: Coordinate of DFNB1 TAD are added on text (line 37) to facilitate understanding.

Round 2

Reviewer 1 Report

English has been improved throughout the document, though there were still a few sentences that are confusing and need rewording:

Lines 14-16: “Non-syndromic 14 hearing loss (i.e., DFNB1) is mostly due to GJB2 variations and DFNB1 large deletions, which remote the gene or cis-regulatory elements (CREs).”

Line 21, remove the s from the end of factors.

Line 42, remove a in front of ubiquitous.

Line 126: “ Indeed, by targeting GC-box, we block suspected that the recruitment of RNA pol II would be blocked and thus GJB2 expression.”

Lines 220-221, need to use either % or fold change, not both.

Other than these, the following minor issues remain:

Line 126, if RNA Pol II recruitment was not tested by ChIP, this should not be mentioned at all. Just say that the GC-box was targeted and prevented transcription.

Mention in the manuscript why it was that you only used MEIS1, and not MEIS2/3.

Author Response

English has been improved throughout the document, though there were still a few sentences that are confusing and need rewording:

Response: We are sorry for these confusions, we had hoped that with English revisions editing services in which we undergone our article it would be better. We have modified sentences that you have pointed. 

Lines 14-16: “Non-syndromic 14 hearing loss (i.e., DFNB1) is mostly due to GJB2 variations and DFNB1 large deletions, which remote the gene or cis-regulatory elements (CREs).”

Response: we added a sentence to better explain this sentence. “Non-syndromic hearing loss (i.e., DFNB1) is mostly due to GJB2 variations and DFNB1 large deletions.”

Line 21, remove the s from the end of factors.

Response: Thank you, we modified this syntax error.

Line 42, remove a in front of ubiquitous.

Response: Thank you, we modified this syntax error.

Line 126: “ Indeed, by targeting GC-box, we block suspected that the recruitment of RNA pol II would be blocked and thus GJB2 expression.”

Response: The sentence was anyway modified following your minor issue:

Line 118:” Indeed, by targeting GC-box, we prevented GJB2 expression”

Lines 220-221, need to use either % or fold change, not both.

Response: Thank you, It is correct, we modified, and we conserved just the percentage (%). Line 136: “Targeting C3, GJB2 expression decreased by 80 % or increased by 2,5 % for CRISPRi and CRISPRa, respectively (Figure 1E and 1F). “ and also at line 124 “When the GJB2 promoter was targeted (sgRNA_1), GJB2 expression, measured by RT-qPCR, increased by 7000 % (Figure 1F). However, when the C3 region was targeted, only sgRNA_2 allowed to increase GJB2 expression by 2,5 % (Figure 1F).”

Other than these, the following minor issues remain:

Line 126, if RNA Pol II recruitment was not tested by ChIP, this should not be mentioned at all. Just say that the GC-box was targeted and prevented transcription.

Response: Indeed, you are right, we corrected it.” Indeed, by targeting GC-box, we prevented GJB2 expression”

Mention in the manuscript why it was that you only used MEIS1, and not MEIS2/3.

Response: Indeed, you are right, we added sentence to explain why we used only MEIS1. Line 154:  “Following these results, we focused on the MEIS factors and in first instance MEIS1 because on JASPAR database, MEIS factor family presents the same DNA biding motif and we have already a MEIS1 antibody which worked.”

Reviewer 2 Report

Unfortunately, my comments were overall not addressed.

Author Response

Unfortunately, my comments were overall not addressed.

Response: We are very sorry, we hoped we have addressed your comments. Thus, we re-tried to better develop and explain some points in the paper. We hope that it will improve understanding. We already used MPDI editing service for English syntax and grammar, that had significantly corrected the language and style. We agree that maybe some sentences are less precise, so we tried to modify them.

Previews comments of the reviewer:

Major:

~300bp region in the ~1000bp C3 enhancer was targeted by four different sgRNAs (2-5) in CRISPRi and CRISPRa experiments for assessing its potential regulatory role.  While sgRNA_2 showed a considerable activating and repressing capacity, sgRNA_3,4,5 showed only activating capacity.  This variability in the activity of very proximal sgRNAs targets could be technical or biological. In addition, the enhancer region is not limited to the 20bp of the sgRNA_2 sequence. Nevertheless, the authors identified binding motifs of MEIS and STAT5A transcription factors on the 20 bps of the sgRNA_2  sequence.

Semi-quantitative RT-PCR indicated that all 3 members of the MEIS family of transcription factors are expressed in SAEC cells and the authors decided to focus on MEIS1 without explaining why.

New response: Indeed, you are right, we added sentence to explain why we used only MEIS1. Line 141: “Following these results, we decided to initially expand analyses with MEIS factors and focused in first instance only on MEIS1 because on JASPAR database, MEIS factor family presents the same DNA biding motif and we have already a MEIS1 antibody which worked.”

STAT5A was not detected in the Semi-quantitative RT-PCR but it is not clear why other members of the STAT family that recognize the same motif were not tested.

New response: It is correct we did not verify the expression of the other members of STAT family, we started to test with STAT5A because we have primers but when we have seen high expression of MEIS factors, we decided to initially expand analyses with these factors We are agree that it is a good remark and in further investigations it could be interesting to study STAT5B also. Except if you think that it is really necessary to have it now. In this case we will need more than 10 days for revision..At this time, we added a sentence in the results part and discussion to explain it :

Line 141 “Following these results, we decided to initially expand analyses with MEIS factors and focused in first instance only on MEIS1 because on JASPAR database, MEIS factor family presents the same DNA biding motif and we have already a MEIS1 antibody which worked.”

Line 228 “In further investigations, it could be interesting to also study STAT5B expression and binding of others MEIS factors in our region.”

In addition, I could not find the primer sequences that were used to quantify the expression of MEIS factors. 

New response: We agree with the Referee and apologize for not indicating it. As we have already indicated in the previous revision, we added primer sequences on Table S2: line 147 “(B) Endpoint RT-PCR analysis of MEIS1, MEIS2, MEIS3 and STAT5a expression on SAECs (Table S2). And line 397: “Table S2: PCR primer sequences used for analysis of MEIS1, MEIS2, MEIS3 and STAT5A expression”.

The authors need to explain why MEIS rather than STAT factors were selected for further analysis. The proposed explanation (L135-137) “MEIS transcription factors family plays an important role on gene regulation and chromatin dynamics. MEIS factors can interact with components of the transcriptional machinery for efficient gene expression.” Is rather general, and not referenced. It is okay to exclude STATs from further analysis but the choice in MEIS has to be justified.

Previously response: The reviewer is right, we missed to reference these sentences. We added sentences and the reference on the text and on the references paragraph.

Lines 138-145 “The MEIS transcription factor family plays an important role on gene regulation and chromatin dynamics [31]. MEIS factors can interact with components of the transcriptional machinery for efficient gene expression [31,32]. Following these results, we decided to initially expand analyses with MEIS factors and focused in first instance only on MEIS1 because on JASPAR database, MEIS factor family presents the same DNA biding motif and we have already a MEIS1 antibody which worked.”

MEIS1 binding on the C3 enhancer was tested by ChIP-PCR. To argue for MEIS1 binding, the signal at the C3 locus must be normalized to a negative control locus (a region that MEIS1 does not bind) in all the ChIP experiments and positive control (a region where MEIS1 is expected to bind) must be shown. Notably- the depiction of MEIS1 binding does not indicate its function. MEIS1 regulatory function can be tested by its perturbation.

We thought we have addressed this comment in the previous revision, if it is not the case could you clarify your request? “

Previously response: Reviewer comment is correct and very relevant, we analysed two other regions (positive and negative controls) and we modified the legend figure.

(Lines 148-153: “We used a negative control (CTL-) that had already been described as a CTCF binding region [33], and a positive control (CTL+), which was described in [34]. The results of one biological replicate are shown. Full gel pictures and data from two additional replicates can be found in Supplementary Figure S12.

It is correct, we cannot affirm that MEIS1 is implicated on GJB2 regulation, we just can speculate of its binding. We accordingly modified the text:

Lines 157-158 “We confirmed that MEIS1 binds to our C3 region, and seems to be implicated in GJB2 regulation. could regulate GJB2 expression by this interaction.”

And we modify on discussion (lines 235-237): “MEIS1 seems to play a key role in gene expression and thus in for GJB2 expression. It would be useful to disrupt its binding to test its regulatory function on GJB2.” 

The 4C-seq experiments support the claim of a long-range association between GJB2 gene with the C3 region, as well as with the 625kb distant locus. However, the reporter experiments and genomic data in figure 4 do not support that the 625kb distant locus serves as an insulator.

New response: Indeed, we just can speculate that -625kb region could be serve as an insulator. So we modified sentences in the results and discussion: line 224 “Thus, this region at -625 kb from GJB2 may be lead to recruit CTCFs.” And line 258: “Thus, we purpose that a DNA loop at CTCF sites inside the promoter and at -631 kb, could permitted GJB2 regulation by bringing GJB2 CREs closer to the promoter.”

Minor:

Figures- overall the text is too small to read.

New response: We modified all figures: text sizes are increased.

Figure 1D- genomic coordinates should be added.

New response: We added the genomic coordinates of GJB2 promoter (chr13:20766541-20768081) and C3 region (chr13:20993543-20994585) below boxes.

Figure 1F- graphs need titles.

New response: We added a title ‘CRISPRi’ and ‘CRISPRa’ respectively on the bottom of figure 1E and F.

Why two different MEIS1 forward primers are described in the ChIP method?

New response: We agree that these two primers are not necessary and confusing, we modified the primers (Table S5) and ChIP method. Lines 367-368: “PCR were performed in triplicate using the HotStarTaq Master Mix (Qiagen), and the primer sequences used are listed in Table S5.”

Some data that seems trivial (for example Figure 1B, C, E) could be moved to supplementary data. Figure 3 can show only the averaged profiles, leaving the replicas in supplementary data.

New response: We agree with the reviewer that Figure 1E is less important and we moved it to supplementary data, Figure S1. For Figures 1B and C we find interesting to let them to permit to readers to directly see systems used.

For Figure 3, it is a good recommendation. We conserved just one replicate and the median profile on Figure 3. We added the replicates of 4C-seq profiles on Figure S3.

Some of the results are overstated. For example, what is the basis to state that C3 region is a “key” enhancer of GJB2(L125) Although the abstract stated that “we determined involvement of several transcription factors on GJB2 expression.” only MEIS1 was tested.

Previously response: We modified the sentences: lines 127-129 “Results correlate between these two approaches and allowed to confirm C3 region as key an important enhancer of GJB2 in endogenous condition” and line 20-21: ”Moreover, with ChIP-PCR we determined involvement of MEIS1 several transcription factors on GJB2 expression.”

The manuscript requires extensive English editing. Just to name few examples:  We already used MPDI editing service for English syntax and grammar, that had significantly corrected the language and style. We agree that maybe some sentences are less precise, so we tried to modify them.

Not clear- We better investigated the effect of CRISPR perturbation on GJB2 expression by targeting these CREs.

Response: Line 17-18:” We better investigated the endogeneous effect of these CREs with CRISPR disruptions and observed GJB2 expression.”

we realized??  chromatin immunoprecipitation (ChIP)

Response: Line 143-145 “Thus, we performed chromatin immunoprecipitation (ChIP) to determine if MEIS1 binds to the C3 enhancer region.”

GJB2 gene regulation was not well understood

Response: Line 15-17: “Although several GJB2 cis-regulatory elements have been described, GJB2 gene regulation remains not well understood.”

Proper citations are missing in multiple locations. For example, L135: “MEIS transcription factors family plays an important role on gene regulation and chromatin dynamics.  MEIS factors can interact with components of the transcriptional machinery for efficient gene expression.”

New response: Lines 138-141: “The MEIS transcription factor family plays an important role on gene regulation and chromatin dynamics [31]. MEIS factors can interact with components of the transcriptional machinery for efficient gene expression [31,32].